# Time-Varying Formation Tracking for Second Order Multi-Agent Systems: An Experimental Approach for Wheeled Robots

Neftali J. Gonzalez-Yances [1],*, America B. Morales-Diaz [1] and Héctor M. Becerra [2]

1   Centro de Investigación y Estudios Avanzados del Instituto Politécnico Nacional, Unidad Saltillo,
    Robótica y Manufacvtura Avanzada, Saltillo 25903, Coahuila de Zaragoza, Mexico;
    america.morales@cinvestav.edu.mx
2   Centro de Investigación en Matemáticas (CIMAT), Guanajuato 36023, Guanajuato, Mexico;
    hector.becerra@cimat.mx
*   Correspondence: neftali.gonzalez@cinvestav.mx

**Abstract:** In this paper, a time-varying formation tracking protocol for second-order multi-sgent systems (MASs) is presented. The time-varying formation considers translation, rotation, and scaling of the geometric pattern that defines the formation. The control law is simple yet effective, and it is composed of a trajectory tracking control and a consensus control that considers the position and velocity feedback of the connected agents in the MAS. The closed-loop system is asymptotically stable, and this was proved using the Gershgoring's disk theorem. The performance of the protocol was extensively tested in experiments using a dynamic extension of the differential-drive robot model. The protocol was tested for different communication topologies and also dealt with switching topologies. The proposed protocol presented good performance regaring both time-varying formation and topology changes. Moreover, a comparison with an existing controller and with only trajectory tracking control has been provided, thus showing that the proposed protocol preserves the formation for all the tested topologies in a better way.

**Keywords:** second-order systems; consensus; multi-agent system; time-varying formation





## 1. Introduction

Cooperative control problems for multi-agent systems (MASs) have been studied in the last two decades. The success of cooperative control strategies relies on the information that the members of a MAS share among them in order to achieve a global task. The engaging control problem is to design suitable algorithms so that a group of agents converge to a desired position that also implies a formation, an agreement, or even a time-varying formation. The scientific community interest in this subject is that it can be applied in diverse areas, such as cooperative surveillance [1], spacecraft formation [2], the formation of unmanned aerial vehicles (UAVs), and autonomous vehicle coordination, among others. Some of the classical control strategies that have been proposed for formation control include leader–follower, virtual structure, and behavioral-based control [3–8].

The backbone of many distributed formation control schemes has been the consensus theory; over the past ten years, several advances have been made in the consensus control of MASs, and various results were derived [9–14]. Several consensus protocols are based on first-order dynamics; see, for instance, [15–22]. However, the motion equations of several vehicles are often modeled as second-order dynamics. For instance, the model of a wheeled mobile robot can be taken to double integrator dynamics for each position coordinate using the feedback linearization technique.

In [23], consensus protocols were stretched to deal with the formation control problems of second-order MASs using a leader–follower and virtual structure approach. More results on consensus based formation control have been reported in [24–26]. In [27], a second-order algorithm under direct information communication was proposed. Several practical

applications involve source seeking and target enclosing, which comprise forming a desired time-varying formation in MASs such as those composed by UAVs [28]. In [29], a consensus and $H_\infty$-based control for heterogeneous multi-agent systems composed of first-order and second-order integrator agents was proposed, and some numerical simulation results were presented to test this approach.

In the state of the art, it is more common to find time-invariant formation tracking controllers and consensus tracking controllers for double integrator MASs with fixed or switching topologies [30–33]. However, these results are not directly suitable to solve time-varying formation tracking problems, such as, for instance, where the formation must be scaled or rotated. To this end, some results can be found in [28] where a time-varying formation tracking controller for second-order MASs with switching topologies was proposed; a leader–follower approach was applied for UAVs, and the design procedure relied on solving a Riccati equation. In the work of [34], a formation control was designed for heterogenous MASs, which was based on a distributed observer to estimate the leader's state. In [35], a distributed model predictive control consensus strategy was proposed to develop a time-varying formation.

In the work of [36], a collision avoidance controller for time-varying formation tracking was developed, and the authors proved their strategy in an extended differential-drive robot model by using simulations. The authors designed a non-linear controller that uses a leader agent strategy based on a consensus-weighted control and a non-linear tracking control to follow a time-varying trajectory. In [37], a distributed tracking control with obstacle avoidance for unicycle-type robots was proposed; the authors merged hierarchical task-based control and consensus control to follow a time-varying reference. From the authors knowledge, there are few proposed methods to deal with time-varying formation tracking, and these are limited in how they deal with simple trajectories, which only are able to scale the formation.

In this work, a time-varying formation protocol for second order MASs is presented, which is based on a trajectory tracking control and a consensus approach. The time-varying formation considers the translation and rotation of the MAS, as well as the formation expansion and contraction. An extensive experimental evaluation is presented using a group of differential-drive robots for different communication topologies. The closed-loop system was asymptotically stable, and this was proved using the Gershgoring's disk theorem. A dynamic extension of the model of the differential-drive robots was used to control the acceleration, which was integrated to send velocity commands to the robots. The control law is simple yet effective; the combination of trajectory tracking control and consensus control results in an improved accuracy of the formation tracking. The implementation of the control law considers the position and velocity of neighboring robots in the MAS, which are obtained from a computer vision system. The proposed protocol also dealt with switching topologies, i.e., the convergence of the tracking error was achieved despite changes in the communication topology. The experimental evaluation included a comparison with an existing controller and discusses the benefits of using the consensus part of the control law in contrast to only using the tracking control.

The outline of the paper is as follows: in Section 2, the problem statement and the proposed control scheme are presented. The error dynamics are defined in Sections 3 and 4, and the stability analysis is developed. In Section 5, an extensive experimental evaluation is provided, and the paper closes with some conclusions in Section 6.

## 2. Problem Formulation and Proposed Control Scheme

A second-order multi-agent system with $n$ agents can be described by the following double integrator model of the $i$th agent:

$$
\begin{aligned}
\dot{\xi}_i &= \zeta_i, \\
\dot{\zeta}_i &= u_i,
\end{aligned}
\tag{1}
$$

where $\xi_i$ represents the position of the *i*th agent in an *m*-dimensional space, i.e., $\xi_i \in \mathbb{R}^m$; therefore, the velocity of the *i*th agent is $\zeta_i \in \mathbb{R}^m$, and the control input $u_i \in \mathbb{R}^m$ corresponds to the acceleration vector of each agent.

Several dynamic models of robotic systems can be simplified to the form of the model (1) by means of a linealization process, such as manipulators [38] and quadrotors [39]. We are particularly interested in formation control of differential-drive robots (DDRs). The kinematic model of a DDR provides first-order relationships for the design of velocity controllers; however, the dynamic extension [40] of the kinematic model allows the designer to treat the system as second order and propose acceleration controllers. In this case, the proposed controller for system (1) will provide the desired accelerations for the dynamic extension. In practice, most of the experimental platforms receive velocity commands, and a low-level controller executes those commands. Thus, the design of the acceleration controllers requires the integration in time of the computed control signals, which has the advantage of providing some smoothness to the commands that are sent to the robot. This aspect is specially useful to diminish discontinuities when formation control is addressed for switching topologies.

**Problem 1.** *We aim to design n acceleration control inputs $u_i$ for each agent modeled as in (1) to track the desired values ($\xi_{iD}(t)$, $\zeta_{iD}(t)$, and $\dot{\zeta}_{iD}(t)$) that define a trajectory in accordance with a formation between agents, such that the following is accomplished:*

- $\lim_{t\to\infty}[\xi_i(t) - \xi_{iD}(t), \zeta_i(t) - \zeta_{iD}(t)]^T = \mathbf{0}$ *for $i \in \{1, \ldots, n\}$,*
- $\lim_{t\to\infty}(\xi_i(t) - \xi_j(t)) = \alpha_i(t)$ *for $i, j \in \{1, \ldots, n\}, i \neq j$,*

*where $\alpha_i(t)$ is a time-varying inter-agent position between agent i and j that can be computed from the reference trajectory $\xi_{iD}(t)$ of each agent.*

*Notice that $\alpha_i(t)$ allows us to define a time-varying formation that can be scaled and/or rotated according to the desired trajectory $\xi_{iD}(t)$. Thus, the challenge in this problem is to accurately track a formation that can be subject to the scaling and rotation of the geometric pattern defining the formation.*

To solve this problem, we propose the following control law that consists of two components:

$$u_i = u_{iT} + u_{iC}, \tag{2}$$

where $u_{iT}$ is a trajectory tracking controller, and $u_{iC}$ is a consensus-based controller. Considering the desired trajectories $\xi_{iD}$, $\zeta_{iD}$, and $\dot{\zeta}_{iD}$, the tracking controller is defined as follows:

$$u_{iT} = \dot{\zeta}_{iD} - \beta(\zeta_i - \zeta_{iD}) - \alpha(\xi_i - \xi_{iD}), \tag{3}$$

with $\alpha > 0$ and $\beta > 0$ being proportional gains for the position and velocity errors, respectively.

The consensus controller $u_{iC}$ takes into account the connections between the agents. The elements $a_{ij}$ of the adjacency matrix $\mathcal{A}$ describe the connectivity topology. Considering the desired trajectoryies $\xi_{iD}$ and $\zeta_{iD}$, the consensus control law is defined as follows:

$$u_{iC} = -\sum_{j=1}^{n} a_{ij}\big(\gamma\big((\xi_i - \xi_{iD}) - (\xi_j - \xi_{jD})\big) + \delta\big((\zeta_i - \zeta_{iD}) - (\zeta_j - \zeta_{jD})\big)\big). \tag{4}$$

$\gamma$ and $\delta$ are non-negative control gains for the position and velocity consensus, respectively. The control law in (2) aims to maintain a formation with multiple robots while each one follows an individual path; therefore, the position and velocity errors of the current agents $(\xi_i - \xi_{iD})$ and $(\zeta_i - \zeta_{iD})$, as well as the ones of its neighbors $(\xi_j - \xi_{jD})$ and $(\zeta_j - \zeta_{jD})$, must be taken into account.

In the following section, we aim to express the control law in terms of the error of the whole multi-agent system to obtain the conditions that the controller gains ($\alpha$, $\beta$, $\gamma$, and $\delta$) must satisfy to guarantee closed-loop stability.

### 3. Error Dynamics

Since the controller in (2) is dependent on the agent's position, velocity, and acceleration, the first- and second-time derivatives of the error must be obtained. Let the position error of the $i$th agent be defined as follows:

$$e_i = \xi_i - \xi_{iD}. \tag{5}$$

The time derivative of (5) gives the velocity error as follows:

$$\dot{e}_i = \zeta_i - \zeta_{iD}. \tag{6}$$

The time derivative of (6) leads to the acceleration error:

$$\ddot{e}_i = \dot{\zeta}_i - \dot{\zeta}_{iD}. \tag{7}$$

The substitution of (5) and (6) into (2) leads to the following:

$$u_i = \dot{\zeta}_{iD} - \beta \dot{e}_i - \alpha e_i - \sum_{j=1}^{n} a_{ij} \big( \gamma (e_i - e_j) + \delta (\dot{e}_i - \dot{e}_j) \big). \tag{8}$$

Given that the control input $u_i$ is directly assigned to each agent's acceleration $\dot{\zeta}_i$ by the agent's model (1) and by using (7), the second-order dynamics of the error are defined by the following:

$$\ddot{e}_i = -\beta \dot{e}_i - \alpha e_i - \sum_{j=1}^{n} a_{ij} \big( \gamma (e_i - e_j) + \delta (\dot{e}_i - \dot{e}_j) \big). \tag{9}$$

For $m$-dimensional agents, consider the $m$-dimensional vectors $\mathbf{e}_i = [e_{i_1}, e_{i_2}, \ldots, e_{i_m}]^T$, $\dot{\mathbf{e}}_i = [\dot{e}_{i_1}, \dot{e}_{i_2}, \ldots, \dot{e}_{i_m}]^T$, and $\ddot{\mathbf{e}}_i = [\ddot{e}_{i_1}, \ddot{e}_{i_2}, \ldots, \ddot{e}_{i_m}]^T$. A generalization of the Equation (9) for a number of $n$ agents of $m$ dimension is defined by the following:

$$\begin{bmatrix} \dot{\mathbf{e}}_1 \\ \dot{\mathbf{e}}_2 \\ \vdots \\ \dot{\mathbf{e}}_n \\ \ddot{\mathbf{e}}_1 \\ \ddot{\mathbf{e}}_2 \\ \vdots \\ \ddot{\mathbf{e}}_n \end{bmatrix} = \left[ \begin{bmatrix} \mathbf{0} & \mathbf{I}_n \\ -\alpha \mathbf{I}_n - \gamma \mathcal{L} & -\beta \mathbf{I}_n - \delta \mathcal{L} \end{bmatrix} \otimes \mathbf{I}_m \right] \begin{bmatrix} \mathbf{e}_1 \\ \mathbf{e}_2 \\ \vdots \\ \mathbf{e}_n \\ \dot{\mathbf{e}}_1 \\ \dot{\mathbf{e}}_2 \\ \vdots \\ \dot{\mathbf{e}}_n \end{bmatrix}. \tag{10}$$

Let us define $\mathbf{e} = [\mathbf{e}_1^T, \mathbf{e}_2^T, \ldots, \mathbf{e}_n^T]^T$, $\dot{\mathbf{e}} = [\dot{\mathbf{e}}_1^T, \dot{\mathbf{e}}_2^T, \ldots, \dot{\mathbf{e}}_n^T]^T$, and $\ddot{\mathbf{e}} = [\ddot{\mathbf{e}}_1^T, \ddot{\mathbf{e}}_2^T, \ldots, \ddot{\mathbf{e}}_n^T]^T$. According to (9), the second-order dynamics of the whole MAS can be rewritten as follows:

$$\begin{bmatrix} \dot{\mathbf{e}} \\ \ddot{\mathbf{e}} \end{bmatrix} = \underbrace{\left[ \begin{bmatrix} \mathbf{0} & \mathbf{I}_n \\ -\alpha \mathbf{I}_n - \gamma \mathcal{L} & -\beta \mathbf{I}_n - \delta \mathcal{L} \end{bmatrix} \otimes \mathbf{I}_m \right]}_{\mathbf{E}} \begin{bmatrix} \mathbf{e} \\ \dot{\mathbf{e}} \end{bmatrix}, \tag{11}$$

where $\mathcal{L}$ is the Laplacian matrix.

The matrix in (11) is the closed-loop error dynamics matrix $\mathbf{E}$ for the controller in (2).

## 4. Stability Analysis

To conduct a stability analysis, the interest is to verify the eigenvalues of the closed-loop error dynamics matrix $\mathbf{E}$ in (10); therefore,

$$det(\mathbf{E} - \lambda \mathbf{I}_{2nm}) = \left| \left( \begin{bmatrix} \mathbf{0} & \mathbf{I}_n \\ -\alpha \mathbf{I}_n - \gamma \mathcal{L} & -\beta \mathbf{I}_n - \delta \mathcal{L} \end{bmatrix} \otimes \mathbf{I}_m \right) - \lambda \mathbf{I}_{2nm} \right|. \tag{12}$$

Inserting the $\lambda$ terms into the Kronecker product in (12) yields

$$det(\mathbf{E} - \lambda \mathbf{I}_{2nm}) = \left| \begin{bmatrix} \mathbf{0} & \mathbf{I}_n \\ -\alpha \mathbf{I}_n - \gamma \mathcal{L} & -\beta \mathbf{I}_n - \delta \mathcal{L} \end{bmatrix} - \lambda \mathbf{I}_{2n} \otimes \mathbf{I}_m \right|. \tag{13}$$

Including the $\lambda$ terms into the error dynamics matrix in (13) yields

$$det(\mathbf{E} - \lambda \mathbf{I}_{2nm}) = \left| \begin{bmatrix} -\lambda \mathbf{I}_n & \mathbf{I}_n \\ -\alpha \mathbf{I}_n - \gamma \mathcal{L} & -\beta \mathbf{I}_n - \delta \mathcal{L} - \lambda \mathbf{I}_n \end{bmatrix} \otimes \mathbf{I}_m \right|. \tag{14}$$

Using the property of the Kronecker product $det(\mathbf{A} \otimes \mathbf{I}) = det(\mathbf{A})^m$ yields the following:

$$det(\mathbf{E} - \lambda \mathbf{I}_{2nm}) = \left| \begin{matrix} -\lambda \mathbf{I}_n & \mathbf{I}_n \\ -\alpha \mathbf{I}_n - \gamma \mathcal{L} & -\beta \mathbf{I}_n - \delta \mathcal{L} - \lambda \mathbf{I}_n \end{matrix} \right|^m. \tag{15}$$

Developing (15) as the determinant of a 2-by-2 block matrix yields

$$det(\mathbf{E} - \lambda \mathbf{I}_{2nm}) = det((-\lambda \mathbf{I}_n)(-\beta \mathbf{I}_n - \delta \mathcal{L} - \lambda \mathbf{I}_n) - (\mathbf{I}_n)(-\alpha \mathbf{I}_n - \gamma \mathcal{L}))^m. \tag{16}$$

Expression (16) can be simplified as

$$det(A - \lambda \mathbf{I}_{2nm}) = det\left( (\lambda^2 + \lambda \beta + \alpha)(\mathbf{I}_n) + (\lambda \delta + \gamma)\mathcal{L} \right)^m. \tag{17}$$

A property of the Laplacian matrix is

$$det(\lambda \mathbf{I}_n + \mathcal{L}) = \prod_{i=1}^{n} (\lambda - \mu_i), \tag{18}$$

where $\mu_i$ is the *ith* eigenvalue of $-\mathcal{L}$.

By comparing (17) and (18), we have the following:

$$det\left( (\lambda^2 + \lambda \beta + \alpha)(\mathbf{I}_n) + (\lambda \delta + \gamma)\mathcal{L} \right)^m = \prod_{i=1}^{n} [\lambda^2 + \lambda \beta + \alpha - (\lambda \delta + \gamma)\mu_i]^m. \tag{19}$$

Rearranging the terms at the right hand of (19) leads to the quadratic expression

$$det\left( (\lambda^2 + \lambda \beta + \alpha)(\mathbf{I}_n) + (\lambda \delta + \gamma)\mathcal{L} \right)^m = \prod_{i=1}^{n} [\lambda^2 + \lambda(\beta - \delta \mu_i) + (\alpha - \gamma \mu_i)]^m. \tag{20}$$

The eigenvalues of the closed-loop error dynamics matrix $\mathbf{E}$ are given by the following:

$$\lambda_i \pm = \frac{-\beta + \delta \mu_i \pm \sqrt{\beta^2 - 2\beta \delta \mu_i - \delta^2 \mu_i^2 - 4\alpha + 4\gamma \mu_i}}{2}. \tag{21}$$

Given that $\mu_i$ are the eigenvalues of $-\mathcal{L}$, and its diagonal elements are as follows:

$$-\mathcal{L}_{ii} = -\sum_{j=1, j \neq i}^{n} a_{ij}; \tag{22}$$

therefore, the Gershgorin's diagram for the $-\mathcal{L}$ matrix consists of a series of discs (Figure 1) located on the left-hand side of the complex plane, and all of them are tangent to the imaginary axis.

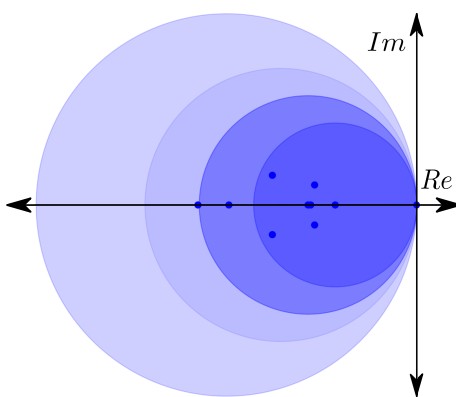

**Figure 1.** Typical distribution of the Gershgorin disks and eigenvalues $\mu_i$ for the $-\mathcal{L}$ matrix.

Due to the location of the Gershgorin's disks, it is correct to consider that the real part of the eigenvalues of $-\mathcal{L}$ is lesser or equal to zero, i.e.,

$$Re(\mu_i) \leq 0. \tag{23}$$

According to (23), the susbtitution of $\mu_i$ for $(-1)|\mu_i|$ in (21) yields the following:

$$\lambda_i \pm = \frac{-\beta - \delta|\mu_i| \pm \sqrt{\beta^2 + 2\beta\delta|\mu_i| - \delta^2|\mu_i|^2 - 4\alpha - 4\gamma|\mu_i|}}{2}. \tag{24}$$

Since the first terms in the numerator of (24) are negative $(-\beta - \delta|\mu_i|)$, to ensure stability, the condition to satisfy is the following:

$$\beta + \delta|\mu_i| > \sqrt{\beta^2 + 2\beta\delta|\mu_i| - \delta^2|\mu_i|^2 - 4\alpha - 4\gamma|\mu_i|}. \tag{25}$$

To satisfy (25), the following condition is obtained:

$$\frac{\delta^2|\mu_i|^2}{2} > -\alpha - \gamma|\mu_i|. \tag{26}$$

The condition (26) is always satisfied given that $\alpha, \beta, \gamma$, and $\delta$ are greater than zero (see Figure 2). Since $\mu_i$ are the eigenvalues of the $-\mathcal{L}$ matrix, they are directly related to the connectivity topology between agents and $|\mu_i| > 0$, which ensures that condition (26) is satisfied under any topology (including the spanning tree). Moreover, for the case of $\mu_i = 0$, the condition (26) becomes

$$0 > -\alpha. \tag{27}$$

Given that $\alpha$ is greater than zero, we can assure that the closed-loop system (11 is stable, even if there are no connections between agents, thanks to the trajectory tracking controller (3).

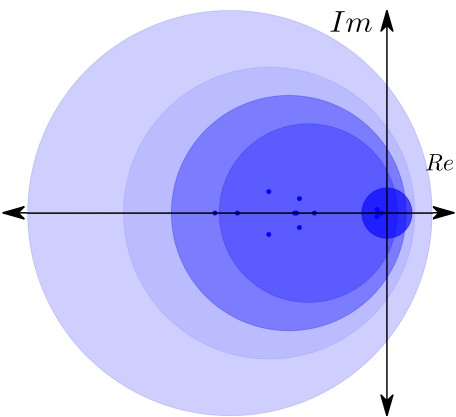

**Figure 2.** Typical distribution of the Gershgorin disks and eigenvalues $\lambda_i$ for matrix **E** in (11).

## 5. Experimental Results

The control protocol (2) was implemented and evaluated for differential-drive robots (DDRs). In order to achieve double integrator dynamics from the kinematics modeling of the DDRs, we uses input–output linearization with dynamic extension [40], and the control law provides accelerations; this is explained in the next subsection. The agent's position is measured with a computer-vision data acquisition system, and the velocity is estimated through the obtained position data.

Differential-drive robots (DDRs)

Consider a differential-drive robot *i* as depicted in Figure 3. The kinematics model of the DDR with dynamic extension is expressed as follows:

$$
\begin{aligned}
v_{x_i} &= \dot{x}_i = v_i \cos \theta_i, \\
v_{y_i} &= \dot{y}_i = v_i \sin \theta_i, \\
\dot{\theta}_i &= \omega_i, \\
\dot{v}_i &= u_i,
\end{aligned}
\tag{28}
$$

where $x_i$ and $y_i$ are the position coordinates of the rear wheels axis center (see Figure 3), $v_i = [v_{x_i}, v_{y_i}]^T$ is the velocity vector of the same point, $\theta_i$ is the angle that denotes the heading of the *DDR* with respect to the *x* axis, $\omega_i$ is the angular velocity, and $u_i$ is the robot's translational acceleration, which is always parallel to the velocity $v_i$.

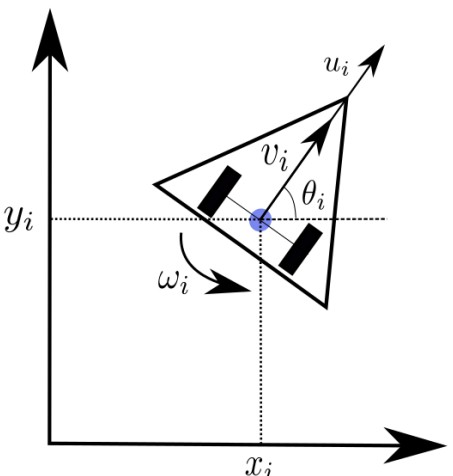

**Figure 3.** DDR model.

To obtain a transformation from the control input ($u_i \in \mathbf{R}^2$) in the Cartesian plane to the robot's translational acceleration ($u_i$) and angular velocity ($\omega_i$), consider the position error of the rear wheels axis center (see Figure 3) as follows:

$$e_i = \begin{bmatrix} x_i - x_{iD} \\ y_i - y_{iD} \end{bmatrix}. \tag{29}$$

Considering the first derivative, we yield:

$$\dot{e}_i = \begin{bmatrix} \dot{x}_i - \dot{x}_{iD} \\ \dot{y}_i - \dot{y}_{iD} \end{bmatrix} = \begin{bmatrix} v_i \cos\theta_i - \dot{x}_{iD} \\ v_i \sin\theta_i - \dot{y}_{iD} \end{bmatrix}, \tag{30}$$

The second derivative is

$$\ddot{e}_i = \begin{bmatrix} u_i \cos\theta_i - v_i\omega_i \cos\theta_i - \ddot{x}_{iD} \\ u_i \sin\theta_i + v_i\omega_i \sin\theta_i - \ddot{y}_{iD} \end{bmatrix}. \tag{31}$$

Rearrangement of the terms after the factorization of the $u_i$ and $\omega_i$ terms leads to the following:

$$\ddot{e}_i = \begin{bmatrix} \cos\theta_i & -v_i \cos\theta_i \\ \sin\theta_i & v_i \sin\theta_i \end{bmatrix} \begin{bmatrix} u_i \\ \omega_i \end{bmatrix} - \begin{bmatrix} \ddot{x}_{iD} \\ \ddot{y}_{iD} \end{bmatrix}. \tag{32}$$

Solving for the transformed control input $[u_i, w_i]^T$ yields

$$\begin{bmatrix} u_i \\ \omega_i \end{bmatrix} = \begin{bmatrix} \cos\theta_i & -v_i \cos\theta_i \\ \sin\theta_i & v_i \sin\theta_i \end{bmatrix}^{-1} \begin{bmatrix} \ddot{x}_i - \ddot{x}_{iD} + \ddot{x}_{iD} \\ \ddot{y}_i - \ddot{y}_{iD} + \ddot{y}_{iD} \end{bmatrix}. \tag{33}$$

Then, the pseudo-kinematic model for the agents is defined as follows:

$$\begin{bmatrix} u_i \\ \omega_i \end{bmatrix} = \begin{bmatrix} \cos(\theta_i) & -v_i \sin\theta_i \\ \sin(\theta_i) & v_i \cos\theta_i \end{bmatrix}^{-1} \begin{bmatrix} \ddot{x}_i \\ \ddot{y}_i \end{bmatrix}. \tag{34}$$

The Equation (34) provides the required relationship to implement the proposed control protocol (2) developed for double integrators that now becomes applicable for DDRs. We can define $[\ddot{x}_i, \ddot{y}_i]^T = [u_{x_i}, u_{y_i}]^T$ as the desired agent acceleration vector of the $i$th robot, and each component is computed using the proposed control law (2). The relation provided in (34) requires that the agent $i$ remains in motion, since the matrix in (34) becomes singular if $v_i = 0$.

Since the DDRs are controlled through the angular velocity of the wheels, the following expressions are used to obtain the left ($\omega_l$) and right ($\omega_r$) wheel velocities from the translational and angular robot velocities:

$$\omega_l = \frac{2v_i - \omega_i L}{2R}, \qquad \omega_r = \frac{2v_i + \omega_i L}{2R}, \tag{35}$$

where the base length or distance between wheels is $L$, and $R$ is the wheel's radius.

Notice that the control signal given by the controller is composed of $[u_i, \omega_i]^T$ to compute the desired velocities for each robot wheel $v_i$ and $w_i$; therefore, $u_i$ must be integrated to obtain $v_i$; this operation acts as a low-pass filter. In addition, notice that, for cases where $0 < v_i << 1$ and the direction of $v_i$ is non-parallel to the vector $[\ddot{x}_i, \ddot{y}_i]^T$, $\omega_i$ will change rapidly, which might degrade for a short time the tracking performance of the controller (2) on the DDRs through the pseudo-kinematic model (34).

The robots used are named MitotianiV1 (from the Nahuatl word dancer), which where designed and built in Cinvestav Saltillo and are depicted in Figure 4. These robots receive angular velocity commands for each wheel via Bluetooth, which is an on-board microcontroller that executes a PID control for each wheel. The wheels angular velocity feedback

signals are obtained from 12 pulse per revolution (ppr) encoders that, in combination with a 78.125:1 gearbox, achieve a 937 ppr.

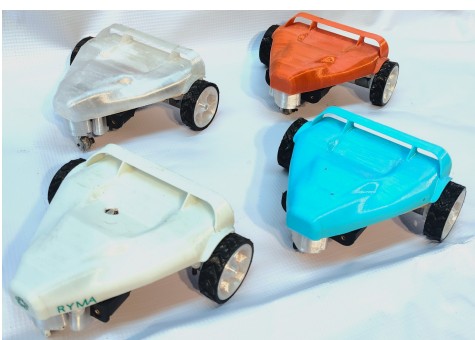

**Figure 4.** Custom-made differential-drive robots used for experimentation.

Data acquisition

To measure every agent's position and orientation, ArUco markers were used in combination with a 1280 × 720 pixel USB camera, which was fixed looking downwards at frame rate of 30 frames per second; the workspace was a plane of 4.13 by 2.32 m; see Figure 5.

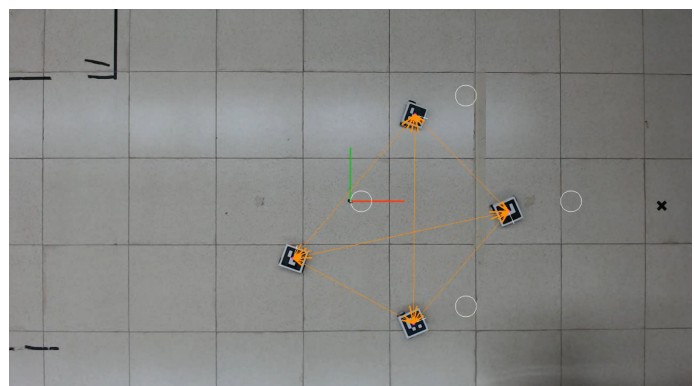

**Figure 5.** Work space provided by the camera's field of view.

*5.1. Experiments*

To test the performance of the controller (2), the task to maintain a complex time-varying formation in the plane is assigned. In the following subsection, the flexible formation is defined. To exhaustively test the properties of (2) under different topologies, several experimental sets are presented. Additionally, (2) is compared with an existing controller and a trajectory tracking controller.

5.1.1. Formation Definition

The controller (2) can be used with any connectivity and formation; to this end a flexible formation is defined in Figure 6 with a virtual center $VC$, the coordinates $(x_c, y_c)$, the angle $\theta_f$ to indicate the orientation of the formation, and the radius $r$. The angle $\phi_i$ defines the desired position of the $i$th agent along a circumference. These parameters provide flexibility to obtain a time-varying formation with changes in translation, rotation, and scale (size).

From this formation definition, the components of the desired trajectory $\xi_D$ for each agent are defined by the following:

$$\xi_{iDx} = x_c + r\cos(\theta + \phi_i).$$
$$\xi_{iDy} = y_c + r\sin(\theta + \phi_i).$$

(36)

Then, the desired velocities in $x$ and $y$ can be obtained considering that $r$, $\theta$, $x_c$, and $y_c$ change along time, and they are are expressed as follows:

$$\dot{\xi}_{iDx} = \dot{x}_c + \dot{r}\cos(\theta + \phi_i) - r\dot{\theta}\sin(\theta + \phi_i).$$
$$\dot{\xi}_{iDy} = \dot{y}_c + \dot{r}\sin(\theta + \phi_i) + r\dot{\theta}\cos(\theta + \phi_i).$$

(37)

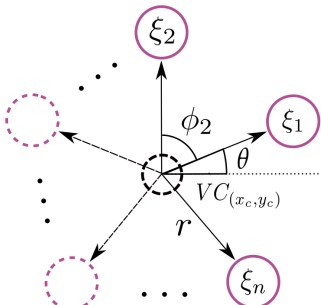

**Figure 6.** Formation definition for $n$ agents.

5.1.2. Evaluation for Different Formation Trajectories

Given that the formation definition allows for translation, rotation, and scaling, the proposed controller (2) was tested with circular trajectories that included formation rotation and scaling.

All the experiments where conducted under three different topologies: fully connected, ring, and spanning tree (see Figure 7). The controller's gains in (2) were determined through the simulation of a circular trajectory. Thirty different values for $\alpha$, $\beta$, $\gamma$, and $\delta$ between the $(0,1)$ interval were tested to simulate all the possible combinations, and eight hundred ten thousand simulations where executed. Only the simulations that saturated the robots wheel velocities ($\omega_{l_{\max}} = \omega_{r_{\max}} = 6.9\frac{rad}{s}$) for less than five hundred milliseconds where considered. The simulation with the lowest error norm summation provided the following controller gains: $\alpha = 0.172$, $\beta = 0.782$, $\gamma = 0.172$, and $\delta = 0.782$, which were selected for experimentation.

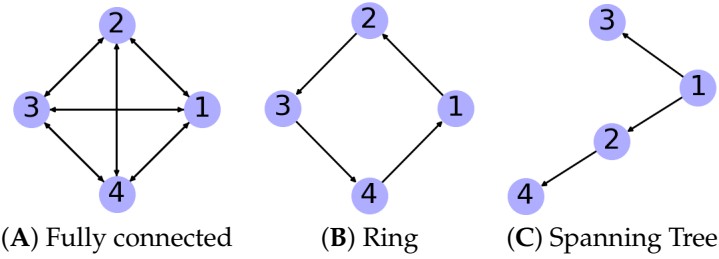

(**A**) Fully connected      (**B**) Ring      (**C**) Spanning Tree

**Figure 7.** Topologies used in the experiments.

In all the experiments, the desired trajectory of the virtual center $VC$ was a circle with a 0.7 m radius and a velocity of 0.1 radians per second, i.e.,

$$x_c = 0.7\cos(0.1t).$$
$$y_c = 0.7\sin(0.1t).$$

(38)

To evaluate the experiments, the average error norm was introduced, and, for a MAS with $n = 4$ agents during an experiment with $p$ samples, it is calculated as follows:

$$\bar{e} = \frac{1}{p}\frac{\sum_{i=1}^{n}\sqrt{(\xi_{ik} - \xi_{iDk})^2}}{n}.$$

(39)

Experiment set 1: Slight rotation.

To generate a demanding trajectory, the formation translation described in (38) was maintained while the formation was slowly rotated, that is,

$$\theta = 0.1(t - t_0).$$
$$r = 0.2. \tag{40}$$

The results for the topologies depicted in Figure 7 are shown in Figures 8–10.

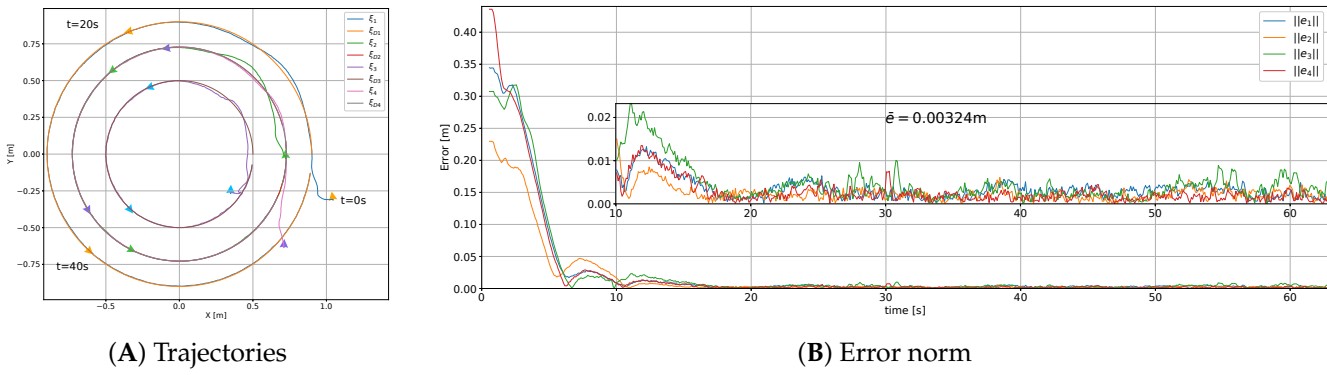

**(A)** Trajectories                                                                              **(B)** Error norm

**Figure 8.** Results for a fully connected topology for experiment set 1; video here: https://cinvestav365-my.sharepoint.com/:v:/g/personal/neftali_gonzalez_cinvestav_mx/ EekX1JAq2WpOi3RJ7lFWkzEBo-FzzWe4zYb_gUoczIlvRA?e=mKQGYw (accessed on 29 June 2023).

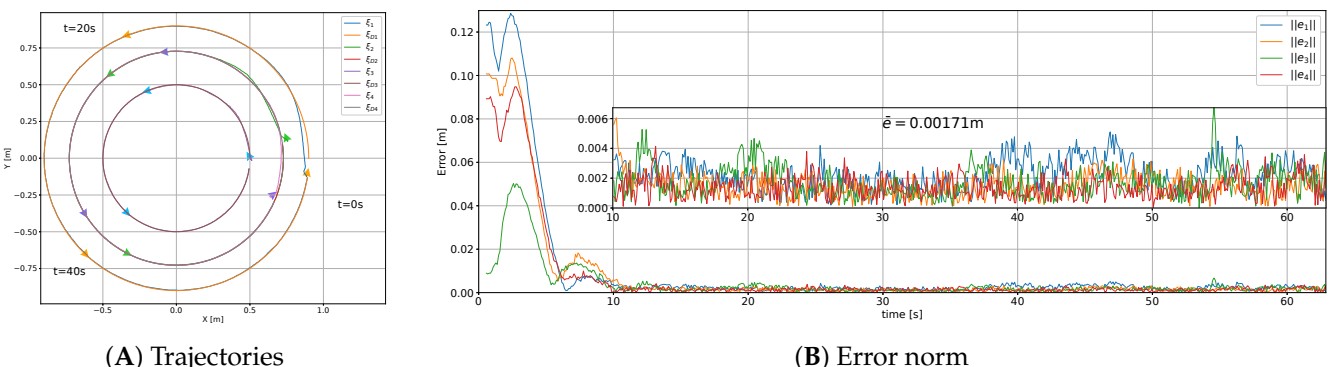

**(A)** Trajectories                                                                              **(B)** Error norm

**Figure 9.** Results for a ring topology for experiment set 1; video here: https://cinvestav365-my. sharepoint.com/:v:/g/personal/neftali_gonzalez_cinvestav_mx/Ee8HmrK47g9HvidnVcbB3 qwB0l3H30CzqcPPdCP5Dw0Rnw?e=hVe0yN (accessed on 29 June 2023).

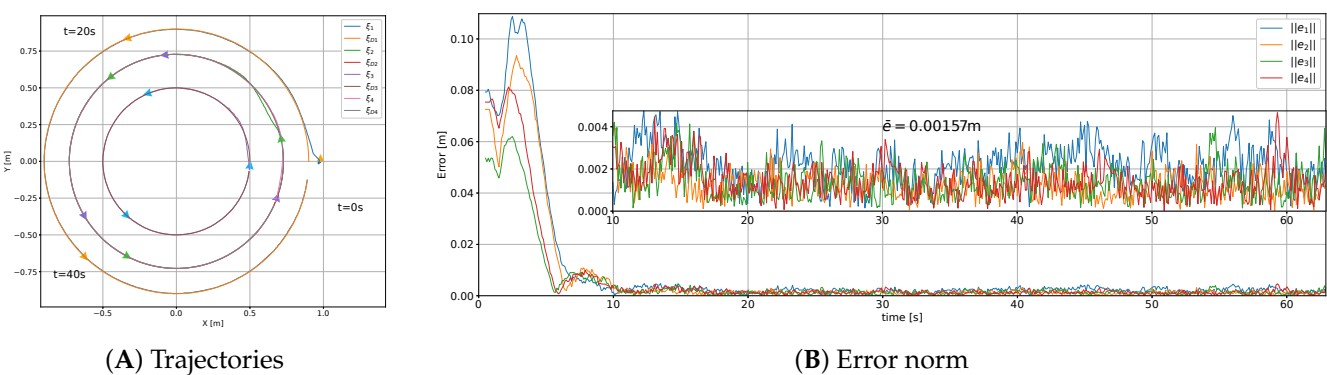

**(A)** Trajectories                                                                              **(B)** Error norm

**Figure 10.** Results for a tree topology for experiment set 1; video here: https://cinvestav365 -my.sharepoint.com/:v:/g/personal/neftali_gonzalez_cinvestav_mx/EaIrWd90HJtBqFKiZ0q_ 0mwBFF1jWs_OrA_4d9i7dysT6w?e=YpvvL2 (accessed on 29 June 2023).

Experiment set 2: Translation, rotation, and scaling.

The purpose of this set of experiments was to test the controller behavior when the formation radius and orientation changed; the formation's orientation and radius are defined as follows:

$$\theta = 0.3(t - t_0).$$
$$r = 0.4 + 0.2cos(0.2(t - t_0)). \tag{41}$$

This case represents a challenging case of time-varying formation. The corresponding results are shown in Figures 11–13.

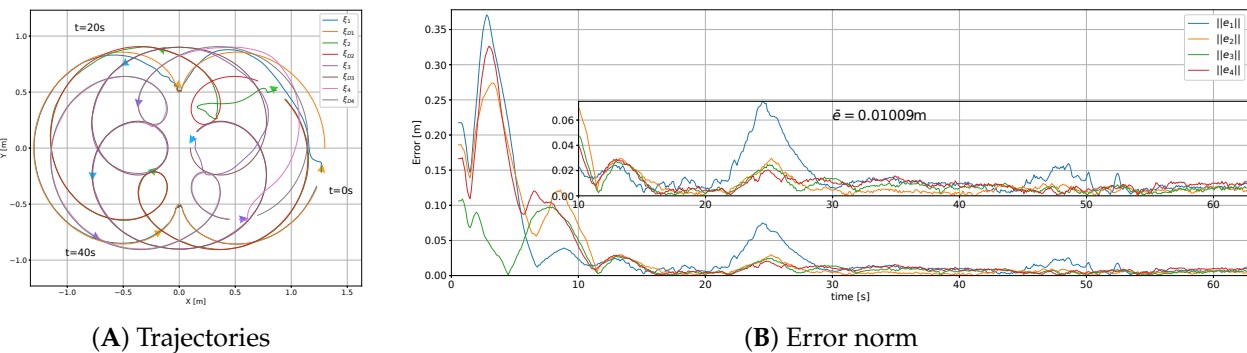

**(A)** Trajectories　　　　　　　　　　　　　　　　　　**(B)** Error norm

**Figure 11.** Results for a fully connected topology for experiment set 2; video here: https://cinvestav365-my.sharepoint.com/:v:/g/personal/neftali_gonzalez_cinvestav_mx/ EbmFh_xM02BLrVyme7jZHS4BGVZmCdeiPMTRf3R2RTUpvA?e=DrIiEI (accessed on 29 June 2023).

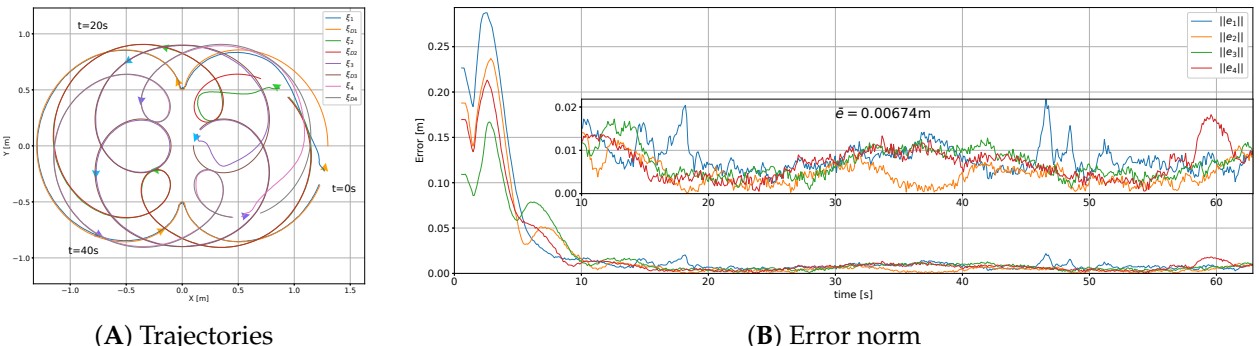

**(A)** Trajectories　　　　　　　　　　　　　　　　　　**(B)** Error norm

**Figure 12.** Results for a ring topology for experiment set 2; video here: https: //cinvestav365-my.sharepoint.com/:v:/g/personal/neftali_gonzalez_cinvestav_mx/EX-W- LE2pZFOnodl238A6kkBk5WPDGM-ilepW0NKDWykTw?e=mIA9aA (accessed on 29 June 2023).

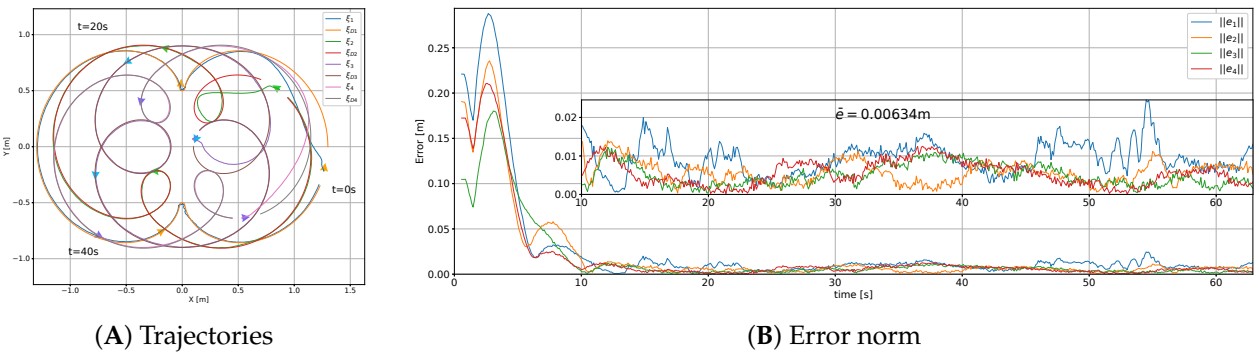

**(A)** Trajectories　　　　　　　　　　　　　　　　　　**(B)** Error norm

**Figure 13.** Results for a tree topology for experiment set 2; video here: https://cinvestav365 -my.sharepoint.com/:v:/g/personal/neftali_gonzalez_cinvestav_mx/EaqroXmsCUFEsPfnhx3R1 1kBp8zU3UrmYJtaLAX_lGPe8Q?e=Luuh9u (accessed on 29 June 2023).

Discussion for experimental sets 1 and 2

According to the results of the formation average error summarized in Table 1, it can be stated that there was a tradeoff between performance and connectivity. Greater connectivity, as in the fully connected topology, implies greater control output, since there are more agents taken into account in the computation of the control signal. However, greater control signals decrease the formation average error norm.

**Table 1.** Formation average error norm for every connectivity in experimental sets 1 and 2.

| Experiment Set | Fully Connected | Ring | Spanning Tree |
|---|---|---|---|
| Experiment set 1 | 3.24 mm | 1.71 mm | 1.57 mm |
| Experiment set 2 | 10.09 mm | 6.74 mm | 6.34 mm |

Given that the robot recieves wheel velocity commands, the acceleration $u_i$ must be integrated to obtain the agent's velocity $v_i$; this process serves as a low-pass filter for every agent´s velocity. Contrarily, in the angular velocity $\omega_i$ of every agent, noise is present, given that no integration is applied (see Figure 14).

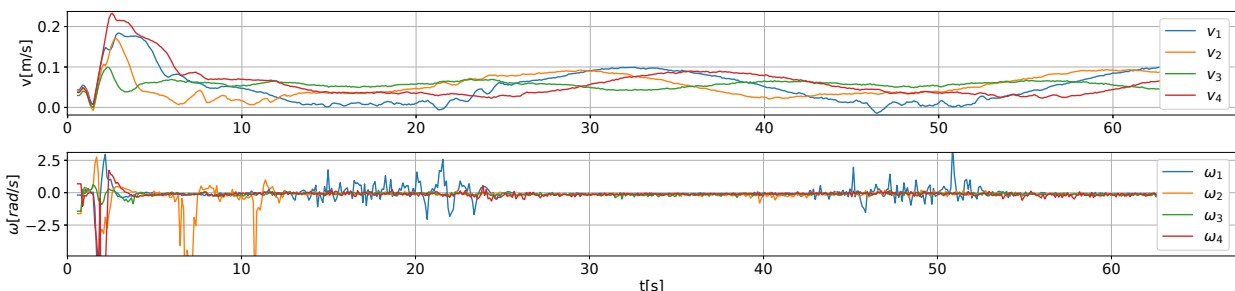

**Figure 14.** Comparison between the agent's translational velocity and angular velocity during experiment 2 with a fully connected trajectory.

The noise from the angular velocity signal can be considered as a perturbation. Higher connectivity implies that this perturbation is transferred in a faster way between the agents. According to this, it can be explained why the fully connected topology has poorer tracking performance than the ring or spanning tree, despite having a greater connectivity and greater control output $u_i$.

5.1.3. Switching Topologies

It is known that consensus-based approaches are able to deal with changes in the communication topologies ([17,28]). In this section, we present the results when the topology is changed among the three considered connectivities of Figure 7.

Experiment set 3: Connectivity changes.

To test the controller's performance when the connectivity between agents changed, the formation's orientation and radius were computed as in (40). The experiment began with a spanning tree topology, which then switched to ring connectivity when the experiment elapsed time (*t*) was 20.94 s, after which the topology changed to fully connected at $t = 41.88$ s. The results are shown in Figure 15.

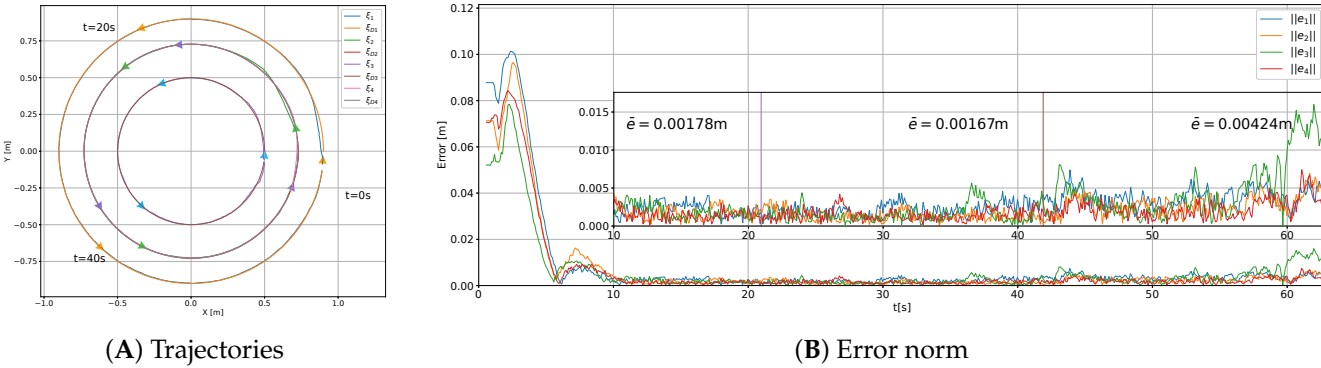

**(A)** Trajectories

**(B)** Error norm

**Figure 15.** Results for the topology change (tree to ring to fully connected) for experiment set 3; video here: https://cinvestav365-my.sharepoint.com/:v:/g/personal/neftali_gonzalez_cinvestav_mx/Eb8VWzzAFnlCkLhshtESV3sBxmuwJEJacR8sGgQ5USr2yA?e=u5Fcr2 (accessed on 29 June 2023).

The effect of the topology change was tested alongside with a perturbation; the topology was switched at $t = 30s$ in combination with the following perturbation in the velocity of agent 1.

$$\dot{\xi}_1 = \begin{cases} 0 & \text{if } 29.5 < t < 30.5 \\ \zeta_1 & \text{otherwise.} \end{cases} \quad (42)$$

The results for perturbation and topology changing from fully connected to ring are shown in Figure 16. To display the results of all possible combinations with the three tested topologies in a compact manner, only the error norm curves in the time interval $[20, 40]$ s are depicted in Figure 17.

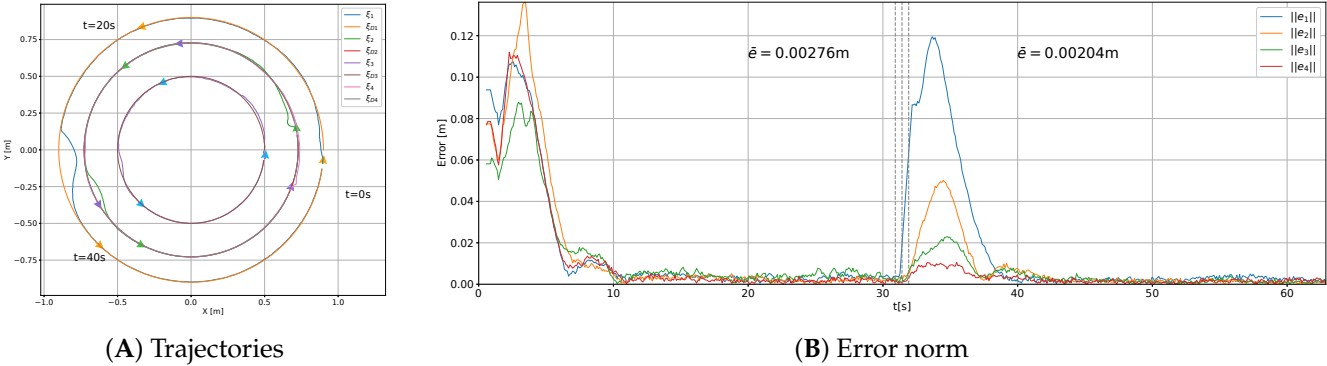

**(A)** Trajectories

**(B)** Error norm

**Figure 16.** Results for the topology change (fully connected to ring) and perturbation for experiment set 3; video here: https://cinvestav365-my.sharepoint.com/:v:/g/personal/neftali_gonzalez_cinvestav_mx/EXIh3GjVC8NOuQye8QP_790BpdkhSIVSNtRyNIPGwephfg?e=cuJ8Xe (accessed on 29 June 2023).

Discussion for topology change experiments

The mean error norm for the experiment of sequential topology change—tree $\rightarrow$ ring $\rightarrow$ fully connected—are depicted in Figure 15 and summarized in Table 2. The increment of the mean error norm in Table 2 is consistent with the results shown in Table 1, since the connectivity was abruptly increased, and the error increased accordingly. The values of error in Table 2 were obtained by considering only the last 10 s of every topology. The sequential increment in the connectivity also implied an increment in the number of links between agents, thereby promoting the transference of rotational velocity perturbation, as was similarly discussed at the end of the previous subsection.

**Table 2.** Mean error norm during the sequential topology change experiment (tree → ring → fully connected).

| Tree | Ring | Fully Connected |
|---|---|---|
| 1.78 mm | 1.67 mm | 4.24 mm |

(**A**) Fully connected to ring

(**B**) Fully connected to tree

(**C**) Ring to fully connected

(**D**) Ring to tree

(**E**) Tree to fully connected

(**F**) Tree to ring

**Figure 17.** Results for the topology change with perturbation for experiment set 3.

The average error norm for the experiments of Figure 17 with simultaneous topology change and perturbation are displayed in Table 3. From the results in Figure 17C,E it is noticeable that the fully connected topology achieved faster perturbation recovery, followed by the ring topology (see Figure 17A,F). Therefore, it can be stated that a low connectivity leads to slower perturbation recovery (see Figures 17D,B).

**Table 3.** Formation average error norm for every connectivity in experiment sets 1 to 5.

| First/Second Topology | Fully Connected | Ring | Spanning Tree |
|---|---|---|---|
| Fully connected | – | 3.13 → 2.24 mm | 3.14 → 1.82 mm |
| Ring | 2.12 → 3.51 mm | – | 1.69 → 1.79 mm |
| Spanning tree | 1.65 → 4.2 mm | 1.9 → 2.31 mm | – |

5.1.4. Comparison with an Existing Controller

The proposed controller (2) was compared with a second-order variant of the controller reported in [41], which is portrayed in (43).

$$
a_i = \frac{1}{k_i} \left( \sum_{j=1}^{n} g_{ij} a_j - \sum_{j=1}^{n} g_{ij} \gamma_0 [(q_i - q_j) - (\delta_i - \delta_j)] + \gamma_i (v_i - v_j) + \right.
$$

$$
\left. g_{ir}[a_r - \gamma_0(q_i - \delta_i - q_r) - \gamma_1(v_i - v_r)] \right),
$$

(43)

where $q_i, v_i$, and $a_i \in \mathbb{R}^2$ are the position, velocity, and acceleration of the agent $i$, respectively, and $\gamma 0$, $\gamma 1$, and $\gamma 2$ are non-negative control gains; $g_{ij}$ are the values in the adjacency matrix that link agent $i$ to agent $j$; $\delta_i$ is the desired distance vector between agent $i$ and the formation center; $g_{ir}$ is a value that defines if an agent has access to the desired position and velocity $(q_r, v_r)$; $k_i$ is defined as $k_i = g_{ir} + \sum_{j=1}^{n} g_{ij}$.

Experiment set 4: Comparison with a similar controller

In this set of experiments, the controller (43) was tested under the spanning tree topology (see Figure 7). For the experiment, a circular trajectory was chosen as in (38) with simultaneous formation rotation, as described in (40). The results are shown in Figure 18.

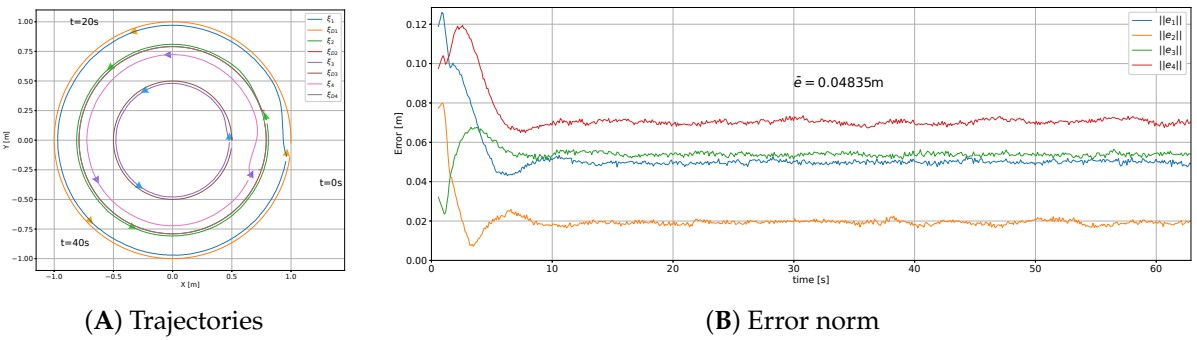

(**A**) Trajectories      (**B**) Error norm

**Figure 18.** Results for the tree topology using an existing controller (43) for experiment set 4; video here: https://cinvestav365-my.sharepoint.com/:v:/g/personal/neftali_gonzalez_cinvestav_mx/ EcRcBnoHVxJJpUuYD2wlMRMByfBLWFvgzdYm3sTuJoDnfA?e=l6S4ck (accessed on 29 June 2023).

To test the controller (43), the desired formation trajectory with translation, rotation, and scaling is defined in (41); the result under the spanning tree topology is displayed in Figure 19.

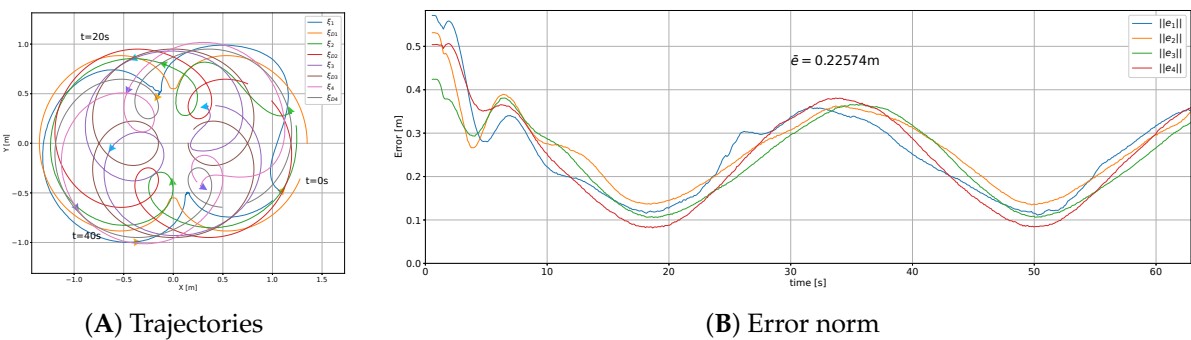

(**A**) Trajectories      (**B**) Error norm

**Figure 19.** Results for the spanning tree topology and the desired formation trajectory (41), using an existing controller (43), for experiment set 4; video here: https://cinvestav365-my.sharepoint.com/:v: /g/personal/neftali_gonzalez_cinvestav_mx/Ef-wE4SLTM5GgjphnvWsEXwBOxz9VsBW3ngx3 lZfIOgGTA?e=ExHzhF (accessed on 29 June 2023).

Discussion for the controller comparison experiments

For the desired formation with a simple trajectory (38), the average error norm in the experiment with the compared controller (43) was 48.35 mm, which was in contrast with the result for the proposed controller (2), which was 1.57 mm. For the complex trajectory with translation, rotation, and scaling (41), the average error norm for the experiments with the compared controller (43) was 22.57 mm, which was in contrast with the result for the proposed controller (2), which was 6.34 mm. This was consequence of how the distance between each agent and the virtual center $\delta_i$ was considered in the compared controller (43). For a formation trajectory where the agents have different velocities or accelerations (i.e., curved formation trajectories or formation, rotation, and scaling), this controller was unable to follow the desired formation trajectory, since the agent´s velocity or acceleration with respect to the formation center was not considered; hence, the proportional behavior of the error with respect to the formation radius, as shown in Figure 19.

### 5.1.5. Trajectory Tracking Comparison

The following experiments are presented to compare the proposed approach in (2), which combines a trajectory tracking term and a consensus control with only trajectory tracking.

Experiment set 5: Trajectory tracking controller comparison.

To compare the performance of the proposed controller (2) with the trajectory tracking term only (3), we considered the more complex case of formation with translation, rotation, and scaling. Then, the formation trajectory in (41) was set as the desired one, and the results are shown in Figure 20.

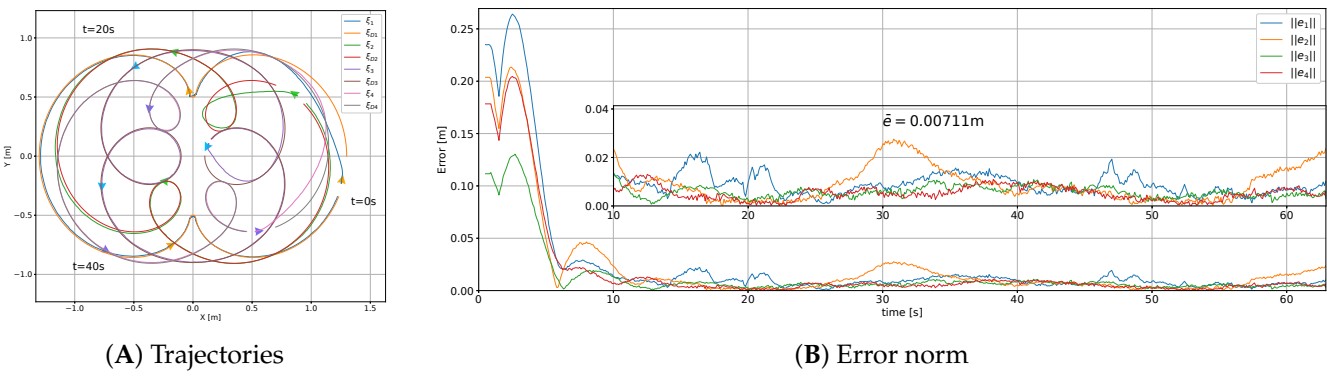

**(A)** Trajectories        **(B)** Error norm

**Figure 20.** Results for the trajectory tracking term only (3) without consensus control for experiment set 5; video here: https://cinvestav365-my.sharepoint.com/:v:/g/personal/neftali_gonzalez_cinvestav_mx/EUHELjVi1ZtPtGqT4waUwKcB-G4GATFyfa5k7fPUGs3gpg?e=lZsIpX (accessed on 29 June 2023).

The results displayed in Figure 20 show that the trajectory tracking controller (3) could follow trajectories with formation translation, rotation, and scaling. This experiment can be compared with the results in Figures 11–13.

The performance of the trajectory tracking controller (3) under the disturbance (42) was tested when the formation was moved on the plane and slowly rotated, as mentioned in (40). The results are depicted in Figure 21. For comparison purposes, the proposed controller (2) was also tested under the same conditions (see Figures 22–24).

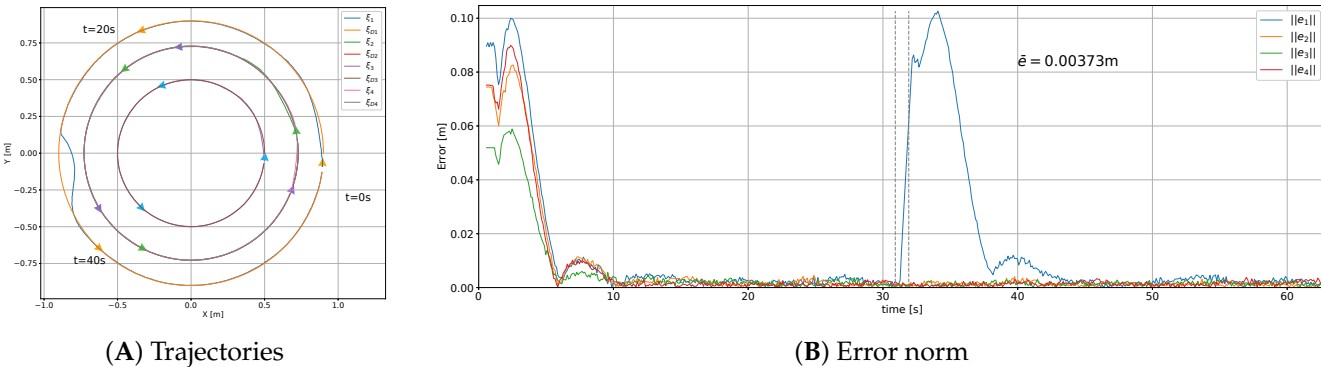

(**A**) Trajectories      (**B**) Error norm

**Figure 21.** Results for the non-consensus control (3) under the disturbance (42) for experiment set 5; the vertical dashed lines indicate the duration of the perturbation (42); video here: https://cinvestav365-my.sharepoint.com/:v:/g/personal/neftali_gonzalez_cinvestav_mx/EZimkegmYQhKuiU2o2OPp58BP2a9z00NWKuSQwAvY-KzUA?e=lf0JSn (accessed on 29 June 2023).

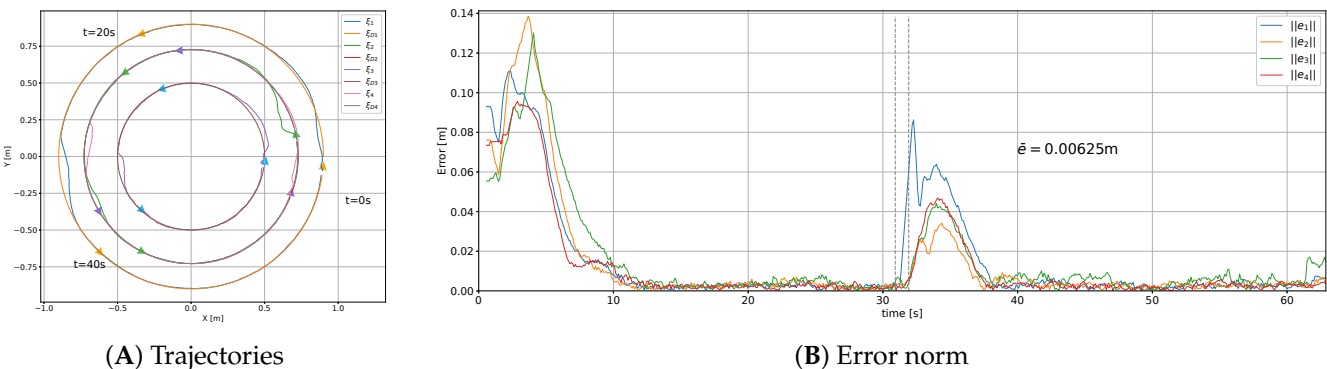

(**A**) Trajectories      (**B**) Error norm

**Figure 22.** Results for the proposed controller (2) under the disturbance (42) and fully connected topology for experiment set 5; the vertical dashed lines indicate the duration of the perturbation (42); video here: https://cinvestav365-my.sharepoint.com/:v:/g/personal/neftali_gonzalez_cinvestav_mx/EcPHrIuWplRAi3fWRd3jFvsBXsb-QH8WbOw5gm_9rPrr4w?e=NmrNYf (accessed on 29 June 2023).

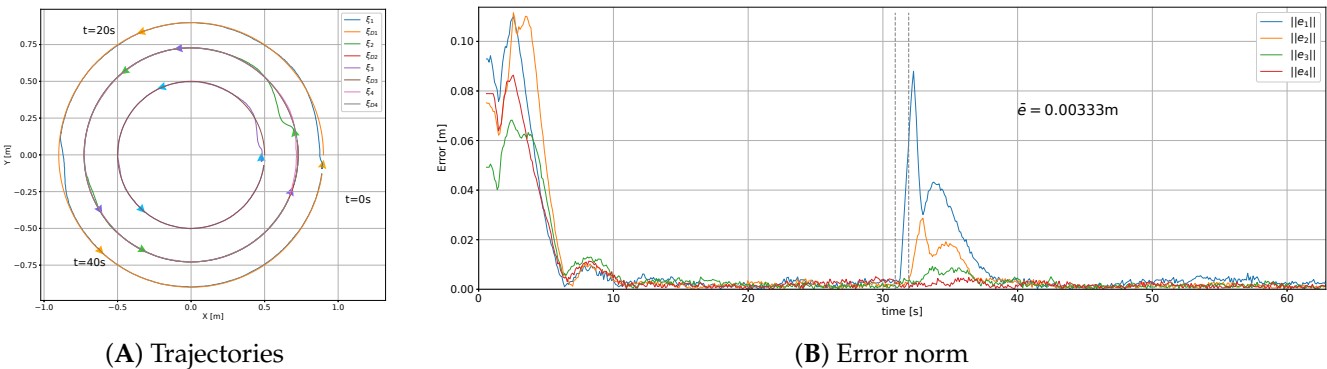

(**A**) Trajectories      (**B**) Error norm

**Figure 23.** Results for the proposed controller (2) under the disturbance (42) and ring topology for experiment set 22; the vertical dashed lines indicate the duration of the perturbation (42); video here: https://cinvestav365-my.sharepoint.com/:v:/g/personal/neftali_gonzalez_cinvestav_mx/EYSmzUAWlgpGiK_sKrCjPooB0Vlh5q5u5wGVrAZNpgc1vA?e=owDBdq (accessed on 29 June 2023).

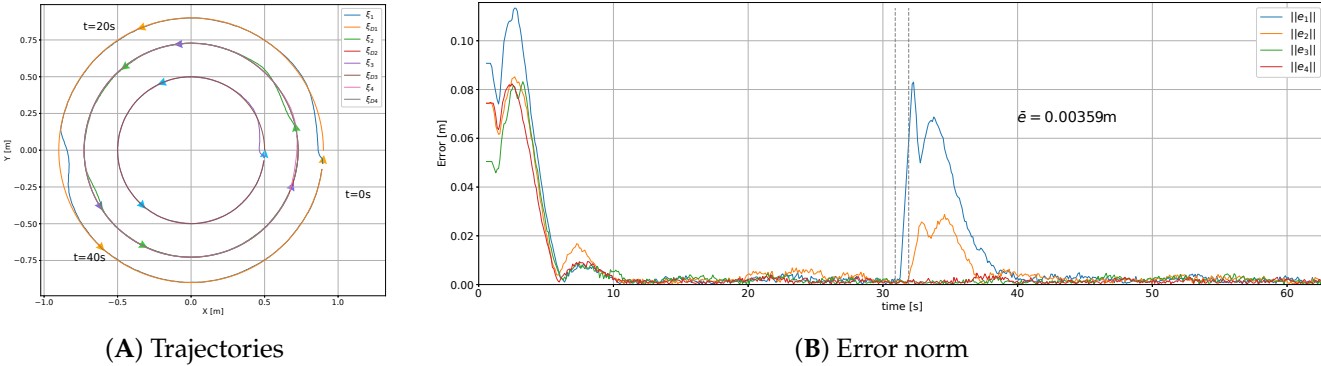

**(A)** Trajectories         **(B)** Error norm

**Figure 24.** Results for the proposed controller (2) under the disturbance (42) and tree topology for experiment set 5; the vertical dashed lines indicate the duration of the perturbation (42); video here: https://cinvestav365-my.sharepoint.com/:v:/g/personal/neftali_gonzalez_cinvestav_mx/EVWI2JeIP7RCka-Gn3uaHJcBMznb0kcSwT691PJbqCegng?e=hzdbdu (accessed on 29 June 2023).

Discussion for the trajectory tracking comparison

As was aforementioned, the experiment for the controller (3) presented in Figure 20 was also carried out with the proposed controller (2) under the three different topologies in experiment set 2 (Figures 11–13). To summarize, the average error norm in those experiments is displayed in Table 4.

**Table 4.** Average error norm for the trajectory tracking controller for experiment set 5 and the proposed controller for experiment set 2.

| Trajectory Tracking Controller | Fully Connected | Ring | Spanning Tree |
|---|---|---|---|
| 7.11 mm | 10.09 mm | 6.74 mm | 6.34 mm |

According to the average error norm in Table 4, the trajectory tracking controller (3) was ranked between the ring and fully connected topologies for the controller (2). The lowest average error norm was achieved by the controller (2) under the spanning tree topology, and the largest average error norm was obtained by the controller (2) with the fully connected topology.

Regarding the perturbation recovery of the tracking controller (3) in Figure 21, it exhibited a greater recovery time (43.94 s) than the controller (2), (38.8 s for spanning tree connectivity), regardless of the connectivity between the agents.

## 6. Conclusions and Future Work

A controller was proposed to generate the acceleration inputs for a second-order MAS such that all the agents followed a time-varying formation trajectory in a coordinated manner. The convergence of the closed-loop error dynamics was demonstrated through the Gershgorin's circle theorem for $n$ agents of dimension $m$ under any connectivity of communication, and a sufficient stability condition was provided. An extensive experimental evaluation of the controller was carried out with three different topologies: fully connected, ring, and spanning tree. The experimental set consisted of four custom-built differential-drive robots (DDRs) and a data acquisition system based on a common webcam. The four robot MASs were able to follow a time-varying formation that considered translation, rotation, and scaling with good accuracy. We noticed a direct relationship between the connectivity and mean error norm, where a larger connectivity led to larger mean error norm.

A dynamic extension was used to map the computed control signal to the DDRs' linear acceleration and angular velocity values. The linear acceleration was integrated in time to send linear velocity to the robots; the angular velocity was directly sent to the robots and

was a signal with a high frequency component, which can be considered as a disturbance. Topologies with higher connectivity contributed to the transfer of this disturbance; as a result, the fully connected topology exhibited a larger mean error norm with respect to the spanning tree topology. The ring topology provided a tradeoff between the connectivity and tracking error, since the mean error norm was slightly larger for the ring than for the spanning tree, but it was smaller than for the fully connected topology.

The proposed controller under the three different topologies and a trajectory tracking controller were subject to an external perturbation to compare their response. According to the experiments, it was found that the connectivity played an important role in the perturbation recovery time and, in general, in transient periods; the controller with a fully connected topology (higher connectivity) recovered faster than when the spanning tree topology was applied. Moreover, the perturbation recovery time for the trajectory tracking controller is larger than the proposed controller under the spanning tree topology.

The proposed controller was challenged by switching the topologies in combination with a perturbation. It was found that, despite having sudden topology changes, neither sharp nor large changes in the error norm signals occurred. Therefore, it can be stated that the proposed controller is able to follow the desired formation trajectory, even with switching topologies. This property can be exploited to reconfigure the consensus component of the proposed controller to achieve a lower mean error norm or faster perturbation recovery.

The proposed controller was compared with a similar existing controller; the results show that the time-varying formation trajectory tracking task was non trivial, since the compared controller did not converge to the desired formation trajectory, thus exhibiting a mean error norm that was 30 times larger than the result for the proposed controller under the spanning tree topology. Moreover, the mean error norm for the existing controller increased in a second comparison experiment where the desired time-varying trajectory included rotation and scaling. The proposed controller showed superiority in the accurate tracking of complex trajectories.

Regarding future work, the experimental results have shown that one of the major drawbacks of using high connectivity in the communication topology for the proposed controller is the disturbance transference. The main source of disturbance in the system is the one produced in the dynamic extension that maps the control acceleration signal to the DDRs' linear acceleration and angular velocity values, since the angular velocity signal has a noise component. To improve the time-varying formation tracking of the proposed controller, a digital filter for the robot's angular velocity might be proposed to mitigate these perturbation effects. Since the proposed controller must work with smooth desired trajectories, we have considered developing a strategy to obtain continuous formation trajectories from discontinuous trajectories; for instance, this might be achieved through linear interpolation. Given the good properties of formation tracking of the proposed controller, it might be extended for the navigation of a formation with capabilities of translation, rotation, and scaling, which could be coupled with a high-level planner to define the trajectories.

**Author Contributions:** Conceptualization, N.J.G.-Y. and A.B.M.-D.; methodology, A.B.M.-D. and H.M.B.; software, N.J.G.-Y.; validation, N.J.G.-Y., A.B.M.-D., and H.M.B.; formal analysis, N.J.G.-Y., A.B.M.-D. and H.M.B.; investigation, N.J.G.-Y., A.B.M.-D. and H.M.B.; resources, A.B.M.-D.; writing—original draft preparation, N.J.G.-Y.; writing—review and editing, N.J.G.-Y., A.B.M.-D. and H.M.B.; visualization, N.J.G.-Y., A.B.M.-D. and H.M.B.; supervision, A.B.M.-D. and H.M.B.; funding acquisition, A.B.M.-D. All authors have read and agreed to the published version of the manuscript.

**Funding:** This research was funded by the Consejo Nacional de Humanidades Ciencias y Tecnologías, grants 1007678 and A1-S-26123.

**Institutional Review Board Statement:** Not applicable.

**Informed Consent Statement:** Not applicable.

**Data Availability Statement:** Not applicable.

**Conflicts of Interest:** The authors declare no conflict of interest.

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
