# Peer review of "Time-Varying Formation Tracking for Second Order Multi-Agent Systems: An Experimental Approach for Wheeled Robots"

_machines, doi:10.3390/machines11080828_

Round 1

Reviewer 1 Report

Question#1:    Several/Many “Gershgorin-based” stability/formation results can be found in the published literature on MAS.  See for example the paper cited as Ref [23]  (Ren, W. Consensus strategies for cooperative control of vehicle formations)  or the 2010 ICRA paper  by M. Franceschelli, A. Gasparri, A. Giua, and G. Ulivi ”Decentralized  Stabilization of Heterogeneous Linear MAS” https://ieeexplore.ieee.org/abstract/document/5509637

Although the Authors provide a literature review on p.2  stating that “…these results are not directly suitable to solve time-varying formation tracking problems, … where the formation must be scaled or rotated…”  they should further elaborate on the differentiation/novelty of their “Gershgorin stability criterion” with other “Gershgorin results”  for MAS.

Question#2:   The Authors are encouraged to elaborate on Eqn.(8) p. 6/29 about the pseudo- kinematic model for the agents.  Is it derived by differentiating Eqn.(27)  and inverting ? If “YES” what about possible singularities?

Question#3  Noise in the angular velocity signal In the Experiment set 2 (Translation, rotation and scaling) starting on p. 9/19 and extending on p.10/19 there is a mention on measurement noise see Line 258 => “…in the angular velocity ωi of every agent, noise is present given that no integration is applied (see Figure 13). The noise from the angular velocity signal can be considered as a perturbationAlthough the experimental results in this test case are acceptable,  the offered Gershgorin-based stability criterion does not/cannot handle noise.   This is true even if “…. a digital filter for the robot’s angular velocity might be proposed to mitigate these perturbation effects…” as stated in the future work subsection Lines 406 – 407.

The Authors should comment/elaborate on this theoretical issue (not on the experiment).

Author Response

Please read the attached file "Response_to_reviewer_1.pdf"

Reviewer 2 Report

This paper proposes a time-varying formation tracking protocol for second-order multi-agent systems. The experiments show the effectiveness of the proposed control protocol. This paper is well organized and has done lots of experiments. However, the following comments can improve the quality of the paper.

1. This paper has proved the stability of the multi-agent systems with the proposed protocol. However, the corresponding theorem should be summarized to tell the readers the stability condition and topology condition.

2. The proposed controller is shown in (2), from which one can see that the two sub-controllers are designed. The first sub-controller is a tracking controller such that the position and velocity can converge to the ideal position and velocity. The second controller is a consensus controller to obtain a ideal formation. However, the ideal position \xi_iD and velocity \zeta_iD can be designed in advance such that the future movement trajectory can also be given in advance. Hence, the author designs the first sub-controller to realize the main goal. Why is the second controller designed?

3. The design usually requires the ideal trajectory is smooth. If the trajectory is non-smooth, or the derivative of velocity does not exist, how to deal this case.

 4. If the second sub-controller can be omitted, the main contributions of the paper are weakened. Only the experiment research is left contribution. 

none

Author Response

Please read the attached file "Response_to_reviewer_2.pdf"

Round 2

Reviewer 2 Report

The authors have answered all the comments. But, there are still several comments.

1- In the stability analysis, the communication topology should be assumed to include a spanning tree. Hence, the proof process is correct.

2- The authors cannot say that the controller is stable. That is wrong. The authors should say the system is stable.

Author Response

please revise the attached pdf file
